



# Spherical Air Mass Factors in One and Two Dimensions with SASKTRAN 1.6.0

Lukas Fehr[1], Chris McLinden[2], Debora Griffin[2], Daniel Zawada[1], Doug Degenstein[1], and Adam Bourassa[1]

[1]Institute of Space and Atmospheric Studies, University of Saskatchewan, Saskatoon, Saskatchewan, Canada
[2]Air Quality Research Division, Environment and Climate Change Canada, Toronto, Ontario, Canada

**Correspondence:** Lukas Fehr (lukas.fehr@usask.ca)

**Abstract.** Air quality measurements from geostationary orbit by the upcoming instrument TEMPO (Tropospheric Emissions: Monitoring of Pollution) will offer an unprecedented view of atmospheric composition over North America. Measurements over Canadian latitudes, however, offer unique challenges: TEMPO's lines of sight are shallower, the sun is lower, and snow cover is more common. All of these factors increase the impact of the sphericity and the horizontal inhomogeneity of the atmosphere on the accuracy of the air quality measurements. Air mass factors encapsulate the complex paths of the measured sunlight, but traditionally they ignore horizontal variability. For the high spatial resolution of modern instruments such as TEMPO, the error due to neglecting horizontal variability is magnified and needs to be characterized. Here we present developments to SASKTRAN, the radiative transfer framework developed at the University of Saskatchewan, to calculate air mass factors in a spherical atmosphere, with and without consideration of horizontal inhomogeneity. Recent upgrades to SASKTRAN include first order spherical corrections for the discrete ordinates method and the capacity to compute air mass factors with the Monte Carlo method. Together with finite difference air mass factors via the successive orders method, this creates a robust framework for computing air mass factors. One dimensional air mass factors from all three methods are compared in detail and are found to be in good agreement. Two dimensional air mass factors are computed with the deterministic successive orders method, demonstrating an alternative for a calculation which would typically be done only with a non-deterministic Monte Carlo method. The two-dimensional air mass factors are used to analyze a simulated scene with TEMPO over the Canadian oil sands. The effect of a sharp horizontal feature in surface albedo and surface $NO_2$ was quantified while varying the distance of the feature from the intended measurement location. Such a feature in the surface albedo or surface $NO_2$ could induce errors on the order of $5$ to $10\%$ at a distance of $50\,km$, and their combination could induce errors on the order of $10\%$ as far as $100\,km$ away.

## 1 Introduction

The application of differential optical absorption spectroscopy (DOAS) (Platt and Stutz, 2008) to space-borne broadband measurements of backscattered ultraviolet-optical sunlight has been used to monitor atmospheric trace gases since the launch of the Global Ozone Monitoring Experiment (GOME) in 1995 (Burrows et al., 1999). A challenging aspect of these measurements



is the presence of complex multiply scattered light paths; uncertainty in the air mass factors (AMFs), which account for these
light paths, is the largest source of error in DOAS retrievals. While the greatest contributions to AMF uncertainty come from
the assumptions related to the observed scene, such as the shape of the absorber vertical profile and the reflectivity of the
surface, the accuracy of the radiative transfer calculations also plays a role.

Accuracy in the radiative transfer becomes more difficult to achieve as the measurement geometry deviates significantly
from the optimal nadir solar backscatter case in which the sun is high and the line of sight is close to vertical. As the sun moves
lower and the line of sight becomes shallower, or equivalently as the solar zenith angle (SZA) and the viewing zenith angle
(VZA) increase, common assumptions such as a plane-parallel, horizontally homogeneous atmosphere begin to break down.
Limited data during winter months at high latitudes motivates pushing the boundary of acceptable SZAs, and large VZAs are
found at the edges of the swath of a pushbroom style instrument in a sun-synchronous orbit, or at the high-latitude extents
of the field of regard of a geostationary instrument. For example, it is hypothesized that inadequate spherical treatment of the
stratospheric AMF could be responsible for underestimated (even negative) tropospheric NO2 VCDs measured by OMI where
the SZA is high (Lorente et al., 2017). In regions of interest such as urban centers, industrial emitters, or forest fires, large
horizontal gradients exist which may introduce errors under the assumption of horizontal homogeneity, especially for localized
measurements. For example, a study by Schwaerzel et al. (Schwaerzel et al., 2020) simulating aircraft measurements of a $NO_2$
plume estimate that failure to account for horizontal structure can lead to VCDs underestimated by as much as $58\%$. Impacts on
satellite-based measurements may also start to become significant given the increased spatial resolution of the new generation
of instruments.

This work is motivated by Canadian interest in Tropospheric Emissions: Monitoring of Pollution (TEMPO) (Zoogman et
al., 2017), an upcoming geostationary ultraviolet-visible spectrometer scheduled for launch in March 2023. TEMPO's view
of Canadian latitudes meets the criteria described above, with large VZAs and SZAs affecting the accuracy of the radiative
transfer. This is magnified during winter when the sun remains low in the sky all day; for example, the northern extent of the
Athabasca oil sands will not see SZAs under $80°$ near winter solstice. Large SZAs have the additional impact of reducing the
measured signal to noise ratio, and measurement sensitivity to the lower atmosphere decreases as the SZA or the VZA increases.
Pervasive snow cover is another complicating factor, contributing significant uncertainty to standard retrieval algorithms due to
its visual similarity to clouds. Snow may also reduce the validity of the assumption of horizontal homogeneity when snow cover
is patchy or when the snow albedo is variable due to different land classifications. A retrieval's sensitivity to such horizontal
variability is increased as the spatial resolution increases.

Here we present developments to SASKTRAN, the radiative transfer framework originally developed for limb scattering
applications at the the University of Saskatchewan, which facilitate the calculation of AMFs for nadir backscatter measurements
for such applications. We present a brief background for the three radiative transfer methods within SASKTRAN - successive
orders, Monte Carlo, and discrete ordinates - including the recent additions of AMF calculations to the Monte Carlo and
spherical corrections to the discrete ordinates. A summary of the theory used to compute AMFs is presented next, followed by
comparisons of standard one-dimensional AMFs computed via the three methods and a two-dimensional case study examining
the error introduced by the assumption of horizontal homogeneity.





## 2 Radiative Transfer

SASKTRAN (Bourassa et al., 2008; Zawada et al., 2015; Dueck et al., 2017) is a radiative transfer framework containing three core methods for solving the radiative transfer equation: HR (high resolution), MC (Monte Carlo), and DO (discrete ordinates). SASKTRAN-HR uses the method of successive orders in a fully spherical atmosphere, and has been used extensively for limb scattering applications, with the primary application being retrievals of ozone (Bognar et al., 2022), nitrogen dioxide (Dubé et al., 2022), and stratospheric aerosol (Rieger et al., 2019) from the Optical Spectrograph and Infrared Imaging

System (OSIRIS) (Llewellyn et al., 2004). SASKTRAN-MC uses the backwards Monte Carlo method in a fully spherical atmosphere, and is primarily used as validation for SASKTRAN-HR. SASKTRAN-DO is a linearized implementation of the discrete ordinates method in a plane-parallel atmosphere similar to VLIDORT (Vector Linearized Discrete Ordinates Radiative Transfer) (Spurr and Christi, 2019), with optional spherical corrections to the incident solar beam and outgoing line of sight. SASKTRAN-DO itself has not been used operationally but the method is widely used for nadir backscatter applications such as

AMF table generation for trace gas retrievals, ozone profile retrievals, and synthetic radiance calculations. For example, VLIDORT is used for ozone profile retrievals for the Ozone Monitoring Instrument (OMI) (Liu et al., 2010), for AMF tables for the Tropospheric Monitoring Instrument (TROPOMI) (Liu et al., 2021), and for all three applications for TEMPO (Zoogman et al., 2017).

SASKTRAN-DO is the fastest method in SASKTRAN, and with spherical corrections it provides enough accuracy for most

nadir-viewing applications, but it is not capable of modelling horizontally inhomogeneities or accounting for the horizontal distribution of the light path. SASKTRAN-HR can model horizontal effects in a fully spherical atmosphere; for example it has been used to perform two-dimensional limb ozone retrievals with the Ozone Mapping and Profiler Suite Limb Profiler (OMPS-LP) (Zawada et al., 2018). Many lines of sight can be evaluated with little extra computational effort, but currently AMFs must be computed with a finite difference approximation which is time consuming for many vertical layers. SASKTRAN-MC can

also model horizontal effects, but it requires long computation times to achieve sufficiently high numerical accuracy, and lines of sight must be considered individually. The analysis presented here is scalar, but all three methods are capable of performing polarized radiative transfer calculations. The following section describes the theory, the key definitions and settings (see Table 1), and the recent developments that are relevant for AMF calculations for each method.

### 2.1 SASKTRAN-HR

The following is the equation of radiative transfer in a form suitable for the method of successive orders. The radiance $I_n(\boldsymbol{r}, \Omega)$ at position $\boldsymbol{r}$ in direction $\Omega$ that has been scattered $n$ times is given by

$$I_n(\boldsymbol{r}, \Omega) = \int_{s_1}^{0} J_n(\boldsymbol{r}_s, \Omega) k(\boldsymbol{r}_s) e^{-\int_s^0 k(\boldsymbol{r}_t)dt} ds + \tilde{I}_n(\boldsymbol{r}_{s_1}, \Omega) e^{-\int_{s_1}^0 k(\boldsymbol{r}_s)ds}, \tag{1}$$

where $k(\boldsymbol{r})$ is the total extinction due to scattering and absorption. When refraction is not considered, the path behind $\boldsymbol{r}$ is parameterized by $\boldsymbol{r}_s \equiv \boldsymbol{r} + s\Omega$ with $s \leq 0$, and $s_1$ is defined such that $\boldsymbol{r}_{s_1}$ lies on the surface or top of atmosphere. The $n^{\text{th}}$

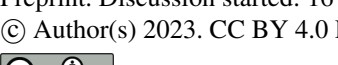



**Table 1.** Summary of SASKTRAN definitions and discretizations for three radiative transfer methods: HR (high resolution, successive orders), MC (Monte Carlo), and DO (discrete ordinates).

| Property | Method | Description | Timing |
|---|---|---|---|
| Diffuse Point | HR | A location where the scattering source term is calculated. Scattering calculations are performed from a chosen set of incoming directions into a chosen set of outgoing directions. | $\mathcal{O}(n)$ |
| Diffuse Profile | HR | A set of diffuse points distributed vertically. One profile is often sufficient, but multiple profiles can be distributed geographically or by local SZA. Additional profiles may be necessary for high SZAs or VZAs. | $\mathcal{O}(n)$ |
| Ray-Tracing Shells | HR and MC | Spherical shells that divide the atmosphere into horizontal layers. Used to divide rays into segments for integration of the extinction and the source terms. | $\mathcal{O}(n^2)$ |
| Photons | MC | Multiply scattered paths traced through the atmosphere. The solar source is sampled at each scatter point. Reaching a maximum number of photons or a specified target accuracy will terminate the calculation. | $\mathcal{O}(n)$ |
| Layers | MC | Optically homogeneous layers in which the radiative transfer equation is solved. Number of layers should increase with total optical depth. | $\mathcal{O}(n)$ |
| Streams | MC | Order of the Legendre polynomial expansion of scattering phase functions and surface reflection functions. | $\mathcal{O}(n^3)$ |
| Optical Heights | All | Vertical grid on which the optical properties of the atmosphere are specified. Linear interpolation is typically used to poll arbitrary heights. | $\mathcal{O}(1)$ |

order source terms accounting for atmospheric scattering $J_n(\boldsymbol{r}, \Omega)$ and surface reflection $\tilde{I}_n(\boldsymbol{r}, \Omega)$ are given by

$$J_n(\boldsymbol{r}, \Omega) = \omega_0(\boldsymbol{r}) \int_{4\pi} I_{n-1}(\boldsymbol{r}, \Omega') \bar{p}(\boldsymbol{r}, \Omega, \Omega') d\Omega' \tag{2a}$$

$$\tilde{I}_n(\boldsymbol{r}, \Omega) = \int_{2\pi} I_{n-1}(\boldsymbol{r}, \Omega') B(\boldsymbol{r}, \Omega, \Omega') \mu(\Omega') d\Omega' \tag{2b}$$

for $n > 1$, where $\omega_0(\boldsymbol{r})$ is the single scatter albedo, $\bar{p}(\boldsymbol{r}, \Omega, \Omega')$ is the scattering phase function, $B(\boldsymbol{r}, \Omega, \Omega')$ is the bidirectional reflectance distribution function (BRDF), and $\mu(\Omega')$ is the cosine of the zenith angle of the incoming direction $\Omega'$. The





formulation is completed by the single scatter source terms:

$$J_1(\boldsymbol{r},\Omega) = \omega_0(\boldsymbol{r})F_0\, e^{-\int_{s_2}^0 k(\boldsymbol{r}+s\Omega_0)ds}p(\boldsymbol{r},\Omega,\Omega_0) \tag{3a}$$

$$\tilde{I}_1(\boldsymbol{r},\Omega) = F_0\, e^{-\int_{s_2}^0 k(\boldsymbol{r}+s\Omega_0)ds}B(\boldsymbol{r},\Omega,\Omega_0)\mu(\Omega_0)\,, \tag{3b}$$

where $F_0$ is the magnitude of the top of atmosphere solar irradiance, $\Omega_0$ is its direction, and $s_2$ is defined such that $\boldsymbol{r}+s_2\Omega_0$ is at top of atmosphere. The diffuse radiation field is solved one order of scatter at a time, using the results from one order to calculate the next.

The radiation field is five-dimensional, with three spatial and two directional coordinates, but due to rotational symmetry around the solar direction the spatial dimensions can be reduced to two (SZA and altitude) when the atmosphere is a function of only altitude and SZA. The radiation field is discretized by selecting so-called diffuse points throughout the atmosphere; locations where the radiance is scattered from a set of incoming directions into a set of outgoing directions. One vertical profile of diffuse points, called a diffuse profile, is often sufficient for scenes with small or moderate zenith angles and a horizontally homogeneous atmosphere. Incoming radiances at diffuse points are computed with explicit ray tracing, dividing the rays into segments according to their intersections with a set of spherical shells, and integrating the extinctions and the source terms. A key advantage of this method is that it computes the full multiple scatter diffuse field, so radiances (and therefore AMF profiles) for any number of lines of sight can be computed with very little extra cost by simply integrating along all lines of sight at the end of the computation. Table 1 summarizes the terminology used to describe the key discretizations used by SASKTRAN-HR.

While SASKTRAN-HR has built-in weighting functions (Zawada et al., 2015), they rely on some approximations which make the result unsuitable for precise AMF calculations, so a finite difference scheme is adopted. More precise placement options for optical properties, ray tracing shells, and diffuse points have been added to facilitate accurate finite difference calculations in one or more dimensions.

## 2.2 SASKTRAN-MC

SASKTRAN-MC applies the backwards Monte Carlo method to the radiative transfer equation separated by order of scatter (Equations 1 through 3), taking random samples of the radiance by explicitly tracing backwards rays that originate at the instrument, are propagated and scattered throughout the atmosphere, and terminate at the sun (Zawada et al., 2015). In simplified notation, the radiance is given by

$$I(\boldsymbol{r},\Omega) = \sum_{n=1}^{\infty}\int_{D_n} f_n(\boldsymbol{x}_n)d\boldsymbol{x}_n\,, \tag{4}$$

where $\boldsymbol{x}_n$ is a parameterization of a light path with $n$ scattering or reflection events, $D_n$ is the space of all such paths ending at position $\boldsymbol{r}$ and direction $\Omega$, and $f_n(\boldsymbol{x}_n)d\boldsymbol{x}_n$ is the radiance contribution from the infinitesimal group of light paths $d\boldsymbol{x}_n$. The radiance and its variance are estimated by Monte Carlo integration: taking samples $\boldsymbol{x}_{nk}$ from probability density function $p_n(\boldsymbol{x}_n)$ via backwards ray tracing, and computing the mean and the variance of $f_n(\boldsymbol{x}_{nk})/p_n(\boldsymbol{x}_{nk})$.

The Monte Carlo method does not rely on the discretization of the diffuse field, and is therefore effective for validating the placement of diffuse points and the choice of incoming and outgoing angular grids in SASKTRAN-HR. As indicated in Table





1, optical heights and ray-tracing shells still need to be chosen. This method is flexible and accurate, and can be run to arbitrary precision, but high precision results require large computation times, and unlike SASKTRAN-HR each line of sight must be considered individually. Therefore it is not feasible for extensive AMF table generation, but it is ideal for validation or for small

studies.

The calculation of box-AMFs and their variances via explicit ray tracing has been recently implemented in SASKTRAN-MC. Further details can be found in Section 3.6.

## 2.3   SASKTRAN-DO

The following is the equation of radiative transfer in a form suitable for the method of discrete ordinates, as developed for

LIDORT (linearized discrete ordinate radiative transfer) in (Spurr et al., 2001):

$$\mu\frac{dI(\tau,\mu,\phi)}{d\tau} = I(\tau,\mu,\phi) - J(\tau,\mu,\phi) \, , \tag{5}$$

where the vertical coordinate $\tau$ is optical depth from the top of the atmosphere, and direction is represented by the absolute value of the zenith cosine $\mu$ and the azimuth $\phi$. The source term $J$ is given by

$$J(\tau,\mu,\phi) = J_{\text{ext}}(\tau,\mu,\phi) + \omega_0(\tau) \int\limits_{4\pi} I(\tau,\Omega')\bar{p}(\tau,\Omega,\Omega')d\Omega' \, , \tag{6}$$

where the first term $J_{\text{ext}}$ consists of thermal emissions and scattering of the direct solar beam, and the second term is the contribution from multiple scattering. The solution to Equations 5 and 6 in a homogeneous slab is computed by expanding the radiance $I$ in a Fourier cosine series in azimuth angle, expanding the phase function $P$ in a series of Legendre polynomials in the cosine of the scatter angle, discretizing $\mu$ by applying Gauss-Legendre quadrature to the integral in the multiple scattering source term, and solving the resulting set of linear first-order differential equations in $\tau$.

SASKTRAN-DO is a separate module within the SASKTRAN framework which uses the discrete ordinates technique to solve the radiative transfer equation in a plane-parallel atmosphere consisting of homogeneous vertical layers. The model is optionally polarized and can calculate analytic derivatives with respect to atmospheric parameters. A pseudo-spherical correction is used which initializes the technique with the solar beam attenuated in a fully spherical atmosphere.

Recently spherical line of sight corrections have been added to SASKTRAN-DO. Here, the single scatter source is cal-

culated exactly in a spherical atmosphere assuming a linear variation in extinction between layer boundaries. The multiple scatter source is approximated by multiple executions of the discrete ordinates technique at a user specified number of solar zenith angles along the line of sight. The observed radiance is then calculated by integrating these source terms in a spherical atmosphere. The spherical mode retains the ability to compute analytic derivatives, but currently is only capable of scalar calculations. The technique is similar to that of the newly released VLIDORT-QS (Spurr et al., 2022). Key parameters controlling

accuracy are described in Table 1.





## 3 Air Mass Factors

The following section presents the theoretical basis for box-AMFs computed with SASKTRAN: through finite difference weighting functions with SASKTRAN-HR, through built-in weighting functions with SASKTRAN-DO, and through explicit ray tracing with SASKTRAN-MC. The traditional framework, based on homogeneous atmospheric layers, is expanded to
allow for alternative vertical discretizations such as the linear interpolation used by SASKTRAN, as well as two- and three-dimensional box-AMFs.

### 3.1 Total AMF

The purpose of the air mass factor (AMF) in DOAS-style retrievals is to transform the slant column density (SCD), a measure of the state of the atmosphere that is heavily coupled with the measurement setup, to the vertical column density (VCD), a
function of atmosphere alone. The AMF is a function of the instrument and sun position, as well as scene information such as surface albedo and cloud cover. For measurements of scattered light, there are a variety of subtle differences between definitions of the SCD and the AMF, depending on different approximations or different variations of the DOAS method. See for example Palmer et al. (Palmer et al., 2001) for one of the earliest popular AMF formulations, Platt and Stutz (Platt and Stutz, 2008) for a comprehensive discussion on DOAS methods, and Rozanov and Rozanov (Rozanov and Rozanov, 2010) for a detailed look
at the subtleties associated with DOAS applied to multiply scattered radiation.

For the following work, the AMF ($A$), the SCD ($S$), and the VCD ($V$) are defined as

$$A \equiv \frac{S}{V} \equiv \frac{\int\limits_L n(l)dl}{\int\limits_0^H n(z)dz},$$

(7)

where $n(z)$ is the number density of the target species, integration over $z$ is along the local vertical from the surface 0 to the top of atmosphere $H$, and integration over $l$ is along the so-called slant path $L$, which is effectively the average path history of
all the light that is captured by the instrument. More specifically, an integral along the slant path is defined here as the radiance-weighted average of the integrals along all contributing light paths. Using the notation of Equation 4, it can be described by

$$S = \int\limits_L n(l)dl \equiv \frac{\sum\limits_{n=1}^{\infty} \int\limits_{D_n} f_n(\boldsymbol{x}_n) \left( \int\limits_{C(\boldsymbol{x}_n)} n(s)ds \right) d\boldsymbol{x}_n}{\sum\limits_{n=1}^{\infty} \int\limits_{D_n} f_n(\boldsymbol{x}_n) d\boldsymbol{x}_n},$$

(8)

where integration over $s$ is along the path $C(\boldsymbol{x}_n)$ represented by parameterization $\boldsymbol{x}_n$.





## 3.2 Continuous AMF

Consider the quantity $dl$ in Equations 7 and 8: it represents the effective length of the average contributing light path within the infinitesimal horizontal layer $dz$. We define the continuous AMF profile,

$$A(z) = \frac{dl}{dz},$$ (9)

describing the enhancement of the slant path compared to the vertical path as a function of altitude. Note that this is now decoupled from the absorber profile $n(z)$ in Equation 7, but still depends weakly on the absorber profile through the radiance contribution $f_n(\boldsymbol{x}_n)d\boldsymbol{x}_n$ in Equation 8. This dependence is typically considered to be negligible under the weak absorber approximation.

AMFs are closely related to derivatives of optical properties, often called weighting functions. Following Rozanov and Rozanov (Rozanov and Rozanov, 2010), the continuous AMF profile $A(z)$ is equivalent (up to a sign) to the functional derivative defined by

$$\int \frac{\partial \ln I}{\partial k}(z)\phi(z)dz = \lim_{\epsilon \to 0} \frac{\ln I[k(z) + \epsilon\phi(z)] - \ln I[k(z)]}{\epsilon},$$ (10)

where $I[k(z)]$ is the measured radiance due to absorber extinction profile $k(z) = n(z)\sigma(z)$, $\sigma(z)$ is the absorption cross section, and $\phi(z)$ is an arbitrary function. This equivalence is evident when linearizing $\ln I[k(z)]$ about a reference profile $\bar{k}(z)$ where $\Delta k(z) = k(z) - \bar{k}(z)$,

$$\ln I[k(z)] = \ln I[\bar{k}(z)] + \int_0^H \frac{\partial \ln I}{\partial k}(z)\Delta k(z)dz + \mathcal{O}(\Delta k^2),$$ (11)

and comparing it to the use of the Beer-Lambert law to describe the difference between these radiances, using the continuous AMF as a change of variables for the slant path integration,

$$\ln I[k(z)] - \ln I[\bar{k}(z)] = -\int_L \Delta k(l)dl = -\int_0^H \Delta k(z)A(z)dz.$$ (12)

This equivalence is convenient; AMFs, which contain information about the distribution of the light path, can be computed from derivatives of radiance with respect to extinction. It is also intuitive, with denser and longer light paths resulting in a larger response from the radiance to a perturbation in extinction.

## 3.3 Discrete AMF

In practice the radiative transfer equation cannot be solved with a continuous vertical coordinate, so the absorber profile $n(z)$, and the rest of the optical properties, must be discretized. We assume the absorber profile $n(z)$ is discretized according to

$$n(z) = \sum_i n_i\phi_i(z),$$ (13)





where the discretizing functions $\phi_i(z)$ are constrained by

$$\sum_i \phi_i(z) = \begin{cases} 1 & 0 \le z \le H \\ 0 & \text{else} \end{cases}, \tag{14}$$

and an effective layer height is defined by

$$\Delta z_i \equiv \int_0^H \phi_i(z)dz. \tag{15}$$

210  $\phi_i(z)$ are typically boxes, corresponding to a model with constant horizontal layers, or triangles, corresponding to a model with linear interpolation on the vertical coordinate.

Using the continuous AMF from Equation 9 as a change of variables, we rewrite the total AMF from Equation 7 as

$$A = \frac{S}{V} = \frac{\int_0^H n(z)A(z)dz}{\int_0^H n(z)dz}, \tag{16}$$

and we plug in the discretized absorber profile from Equation 13, returning

$$A = \frac{\sum_i n_i \int_0^H \phi_i(z)A(z)dz}{\sum_i n_i \int_0^H \phi_i(z)dz}. \tag{17}$$

This motivates the definition of the partial SCD,

$$S_i \equiv n_i \int_0^H \phi_i(z)A(z)dz, \tag{18}$$

the partial VCD,

$$V_i \equiv n_i \int_0^H \phi_i(z)dz = n_i\Delta z_i, \tag{19}$$

220  and the box-AMF,

$$A_i \equiv \frac{S_i}{V_i} = \frac{1}{\Delta z_i} \int_0^H \phi_i(z)A(z)dz. \tag{20}$$



With these definitions, the total AMF is computed according to

$$A = \frac{\sum\limits_i V_i A_i}{\sum\limits_i V_i}\,.$$ (21)

As long as the discretizing functions $\phi_i(z)$ are sufficiently narrow, the box-AMFs $A_i$ are insensitive to the absorber profile, allowing them to be tabulated and used for scenes with arbitrary absorber profiles $V_i$.

### 3.4 Multi-dimensional AMF

This framework is also compatible with two- or three-dimensional box-AMFs, which characterize the horizontal distribution of the measured light path in addition to the vertical. The two-dimensional plane-parallel formulation using Cartesian coordinates will be presented as an example, but similar formulations could be made for three dimensions, or for two or three dimensions in a spherical atmosphere using polar or spherical coordinates. Adding one horizontal coordinate $x$ in a plane-parallel atmosphere, the SCD is computed by

$$S = \int\limits_{-\infty}^{\infty} \int\limits_0^H n(z,x)A(z,x)dz\,dx\,,$$ (22)

where $A(z,x)dz\,dx$ is the length of the slant path that is contained within the area $dz\,dx$. Note that the integral in $x$ is finite due to $A(z,x)$, which approaches zero as you move away from the line of sight. We assume the discretization

$$n(z,x) = \sum_{i,j} n_i^j \phi_i(z)\psi^j(x)\,,$$ (23)

where the horizontal shape functions $\psi^j(x)$ have effective width

$$\Delta x^j \equiv \int \psi^j(x)dx\,,$$ (24)

and are constrained by

$$\sum_j \psi^j(x) = 1$$ (25)

for all $x$. We then define the partial SCD,

$$S_i^j \equiv n_i^j \iint \phi_i(z)\psi^j(x)A(z,x)dz\,dx\,.$$ (26)

the partial VCD,

$$V_i^j \equiv n_i^j \Delta z_i\,,$$ (27)

and the box-AMF,

$$A_i^j \equiv \frac{S_i^j}{V_i^j} = \frac{1}{\Delta z_i} \iint \phi_i(z)\psi^j(x)A(z,x)dz\,dx\,.$$ (28)





Defining the total VCD corresponding to horizontal shape function $\psi^j(x)$,

$$V^j \equiv \sum_i n_i^j \Delta z_i,\tag{29}$$

we seek the total AMF $A^j$ that would be used to retrieve $V^j$:

$$A^j \equiv \frac{S}{V^j} = \frac{\displaystyle\sum_{i,j'} V_i^{j'} A_i^{j'}}{\displaystyle\sum_i V_i^j}.\tag{30}$$

### 3.5 AMFs Via Weighting Functions

Here we derive the relation between box-AMFs and discrete weighting functions. Using the equality of continuous weighting functions and AMF profiles (Equations 11 and 12) the box-AMF (Equation 20) can be written as

$$A_i = \frac{S_i}{V_i} = -\frac{1}{\Delta z_i} \int_0^H \phi_i(z) \frac{\partial \ln I}{\partial k}(z) dz.\tag{31}$$

Applying the functional derivative definition (see Equation 10), we get

$$A_i = -\frac{1}{\Delta z_i} \lim_{\Delta k \to 0} \frac{\ln I[k(z) + \Delta k \phi_i(z)] - \ln I[k(z)]}{\Delta k},\tag{32}$$

which is nearly equivalent to

$$A_i = -\frac{1}{\Delta z_i} \lim_{\Delta n \to 0} \frac{\ln I[n(z) + \Delta n \phi_i(z)] - \ln I[n(z)]}{\sigma_i \Delta n},\tag{33}$$

where $\sigma_i$ is the absorption cross section corresponding to $\phi_i(z)$. The difference between Equations 32 and 33 is a slight change in perturbation shape due to the vertical structure of $\sigma(z)$ which varies with temperature; this change is negligible due to the small vertical gradient of $\sigma(z)$ and the narrow width of $\phi_i(z)$. In the constant layer case, there is no change. Applying a perturbation with shape $\phi_i(z)$ to the profile $n(z)$ is equivalent to perturbing the parameter $n_i$. Therefore the limit in Equation 33 can be rewritten as the following derivative,

$$A_i = -\frac{1}{\sigma_i \Delta z_i I} \frac{\partial I}{\partial n_i}\tag{34}$$

which now contains the derivative of radiance with respect to number density grid points, which is the form of the weighting functions returned by SASKTRAN-HR and SASKTRAN-DO. Two dimensional box-AMFs in SASKTRAN-HR are similarly computed according to

$$A_i^j = -\frac{1}{\Delta z_i \sigma_i^j I} \frac{\partial I}{\partial n_i^j}.\tag{35}$$

These results were found to remain insensitive to a variety of perturbation shapes. For example, using a rectangular $\phi_i(z)$ to compute $V_i$, which is how it is typically defined in the literature, while using a triangular perturbation to compute $A_i$,





which is more compatible with SASKTRAN, was found to still produce accurate results. Some of the early tables discussed in Section 5 employed this strategy, using a triangular perturbation contained within each AMF layer to compute the box-AMF. This required two ray tracing shells and two diffuse points per layer, which drove up the computation time significantly when additional AMF layers were required due to the $\mathcal{O}(n^2)$ dependence on the number of ray tracing shells. This motivated the use of a constant-layer atmosphere representation, which permitted the use of rectangular perturbations, requiring only one ray-

tracing shell and one diffuse point per layer. This strategy resulted in negligible changes to the box-AMFs, and was therefore used for the SASKTRAN-HR box-AMFs presented in Section 4.

### 3.6 Via Ray Tracing

A ray tracing method for computing box-AMFs with SASKTRAN-MC was implemented to be used as validation for the weighting function AMFs. Consider the partial SCD definition in Equation 18; if the slant path integration $A(z)dz$ is replaced

with the definition from Equation 8, the partial SCD becomes

$$
S_i = \frac{\sum_{n=1}^{\infty} \int_{D_n} f_n(\boldsymbol{x}_n)\left(n_i \int_{C(\boldsymbol{x}_n)} \phi_i(s)ds\right) d\boldsymbol{x}_n}{\sum_{n=1}^{\infty} \int_{D_n} f_n(\boldsymbol{x}_n) d\boldsymbol{x}_n} .
$$

(36)

As described in Section 2.2, the radiance and its variance are computed by sampling $\boldsymbol{x}_n$ via backwards ray tracing. To calculate the partial SCD $S_i$ (and therefore the box-AMF $A_i$), the same ray tracing is used to simultaneously estimate the integrals in the denominator (the radiance, as before) and the numerator of Equation 36, explicitly integrating the number

density along each traced light path. The variance of $A_i$ is estimated by computing the variance and covariance of the two integrals, then using a first order Taylor expansion to approximate the variance of their ratio.

## 4 AMF Comparisons

### 4.1 SASKTRAN-MC vs SASKTRAN-HR

The following section presents a series of comparisons between box-AMF profiles generated using SASKTRAN-HR and

SASKTRAN-MC. The 1976 US Standard Atmosphere (Dubin et al., 1976) was used for air density, temperature, pressure, and ozone density profiles, and a typical $NO_2$ density profile was taken from a one year global tropospheric chemistry simulation performed using the Goddard Earth Observing System Model version 5 Earth system model (GEOS-5 ESM) with the GEOS-Chem chemical module (G5NR-chem) (Hu et al., 2018). Computations were performed at $440\,nm$, a typical value for AMFs for $NO_2$ retrievals, and box-AMF layers of thickness $500\,m$ were defined up to $50\,km$.

In Figure 1, the SASKTRAN-HR box-AMFs were computed for moderate geometries, with a low SZA and a typical range of VZAs, first with default settings and then with increased angular resolution for the diffuse field. For both computations,



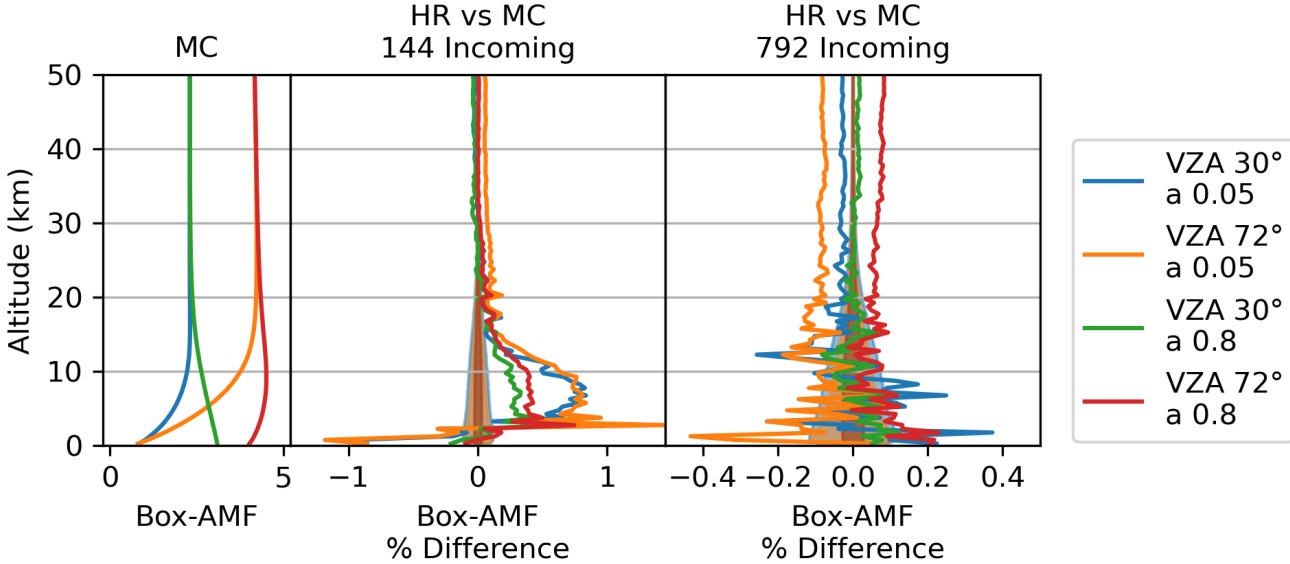

**Figure 1.** Box-AMF comparison between SASKTRAN-MC and SASKTRAN-HR with a SZA of $30°$. The shaded region shows the uncertainty in the SASKTRAN-MC box-AMFs after $10^7$ traced photon paths.

ray tracing for the first $50\,\mathrm{km}$ matched the box-AMF layer boundaries with $500\,\mathrm{m}$ spacing; above that a spacing of $1\,\mathrm{km}$ was used. A single diffuse profile was used, with one diffuse point placed just above the surface, and one diffuse point placed in the center of each ray tracing layer. Typical settings were selected for the first computation: $144$ incoming directions per diffuse point, consisting of $6$ azimuthal directions multiplied by $24$ zenith angles, with $6$ downward facing, $8$ near the horizon, and $10$ upward facing. The result agrees with SASKTRAN-MC within just over $1\,\%$. Calculations were done at surface albedos of $0.05$ and $0.8$; larger percent errors are observed at the lower albedo in Figure 1, but this is a consequence of lower AMFs, not higher errors.

Inadequate resolution in the downward and horizontal incoming diffuse field was found to be responsible for most of the discrepancy. In the second computation, the agreement was brought down to within $0.4\,\%$ by selecting $792$ incoming directions; $12$ azimuthal directions and $66$ zenith angles, with $24$ downward facing, $32$ near the horizon, and $10$ upward facing.

At more extreme geometries, large errors are introduced by the use of a single diffuse profile, due to the larger range of SZAs seen along the long, shallow lines of sight. Using $9$ diffuse profiles, spanning this range of SZAs, was sufficient to bring about the reduction in difference seen in Figure 2. Note that these comparisons are done assuming horizontal homogeneity in the atmosphere; the multiple diffuse profiles are accounting for geometric effects, not atmospheric effects such as photochemical changes in $NO_2$ with SZA. The remaining difference does not appear to respond to increases in HR resolutions, which are already approaching their practical limit. It is perhaps a limitation of the finite difference approximation, or some subtle difference between method-specific configurations which is amplified by the long light paths in the most extreme geometries.



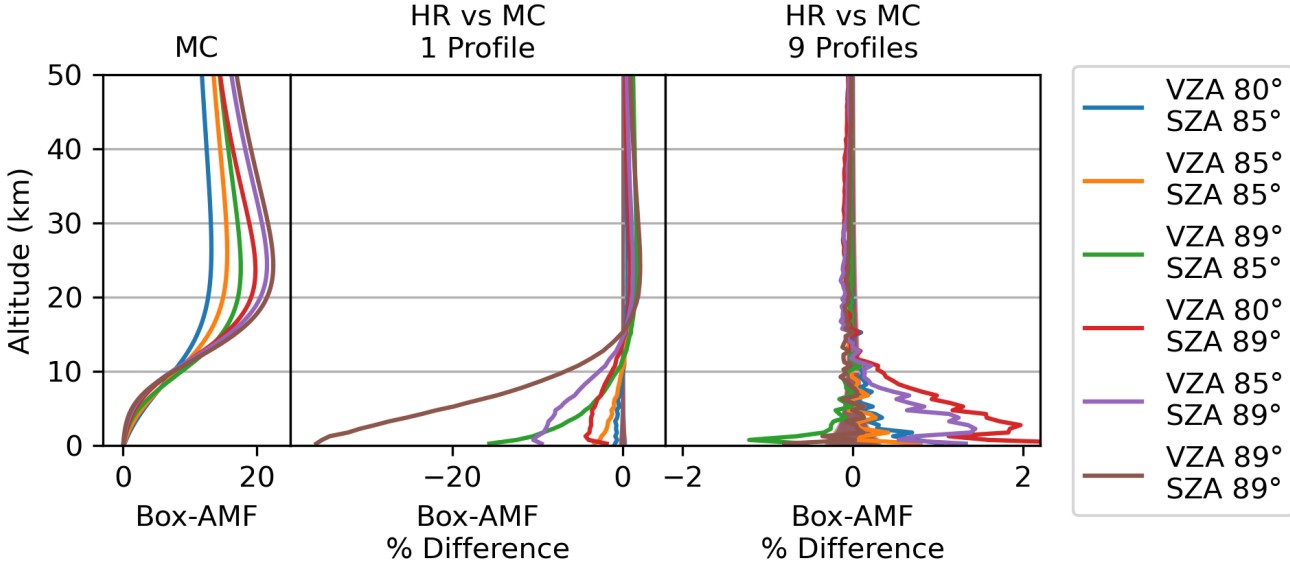

**Figure 2.** Box-AMF comparison between SASKTRAN-MC and SASKTRAN-HR at extreme geometries with the sun low and with shallow lines of sight. The albedo is low (0.05), but the results here are insensitive to albedo, and look similar with a high albedo.

Results for multiple wavelengths spanning a large range of geometries are presented in Figure 3. Wavelengths were chosen

to match common retrievals spanning a wide range of the visible spectrum, with $330\,\mathrm{nm}$ which is typical for formaldehyde retrievals, $440\,\mathrm{nm}$ for $NO_2$, and $600\,\mathrm{nm}$ for ozone retrievals using the Chappuis absorption band. The phenomena explored in Figures 1 and 2 are visible; increasing the diffuse resolution is seen to improve agreement for small to moderate zenith angles, and adding diffuse profiles shows improved agreement for larger zenith angles. The effect of adding diffuse profiles clearly varies with wavelength; at $600\,\mathrm{nm}$ where multiple scattering is less important, the difference is reduced, but at $330\,\mathrm{nm}$ multiple

profiles are shown to be necessary for a larger range of SZAs and VZAs. Even with the increase in diffuse profiles and incoming directions, discrepancies greater than $2\,\%$ remain for SZA $89°$ at $330\,\mathrm{nm}$ and $440\,\mathrm{nm}$. This is due to the proximity of the solar terminator to the ground pixel at such an extreme SZA, in combination with strong contributions from multiple scattering. Under these conditions the solar transmission table, which tabulates the intensity of the direct solar beam as a function of altitude and SZA, would require higher resolutions to accurately capture this discontinuity.

## 4.2   SASKTRAN-MC vs SASKTRAN-DO

The following section examines the accuracy of the pseudo-spherical discrete ordinates solution under the same set of conditions. The following computations used 16 streams in the full space and divide the atmosphere into $250\,\mathrm{m}$ layers. The spherical line of sight correction computes the diffuse field at 5 SZAs along the line of sight. The results are displayed in Figure 4.



**Figure 3.** Box-AMF comparisons between SASKTRAN-MC and SASKTRAN-HR. Transparency is set according to the SZA; fully opaque lines have the minimum SZA of $30°$, and the most transparent lines have the maximum SZA of $89°$.





**Figure 4.** Box-AMF comparisons between SASKTRAN-MC and SASKTRAN-DO. Transparency is set according to the SZA; fully opaque lines have the minimum SZA of $30°$, and the most transparent lines have the maximum SZA of $89°$. Note that the x-axis scale changes in the rightmost column.





The effect of the solar spherical correction in Figure 4 is subtle but visible, correcting cases with low VZA and high SZA

(see the blue and green transparent lines in the leftmost column compared to the middle). The addition of a spherical line of sight correction dramatically improves cases with large VZAs. With these two corrections, the discrepancy is brought to within roughly $3\%$. This discrepancy is only weakly dependent on geometry, and what little dependence there is has been reversed, with higher VZAs resulting in smaller discrepancies. Therefore, if the uncorrected plane-parallel solution is adequate at moderate geometries for a given application, the line of sight corrected solution should be considered adequate at all geometries for

that application.

There is a distinct $1\%$ to $3\%$ feature in the middle altitudes which persists even at small zenith angles. This feature is insensitive to the number of streams, layers, and discrete SZAs at which the diffuse field is computed. The peak of the feature, which descends as the atmosphere becomes more transparent at the higher wavelengths, shows the altitude where significant multiple scattering paths reside; below the effect is suppressed by higher optical depths along longer paths, and above it is

suppressed by lower scattering extinction. Note that this feature is absent from HR and MC as they do not assume a plane parallel atmosphere for multiple scattering.

To test if this difference is due to the plane-parallel assumption, we repeat the calculation in a less spherical atmosphere. This effective flattening was not achieved by changing the radius of the Earth within SASKTRAN, rather an equivalent effect was produced by reducing the vertical scale of the atmosphere by a factor of 10 and increasing all scattering and absorbing

concentrations by a factor of 10. The results, shown in Figure 5, show that the flattened atmosphere reduces the feature. A new error feature is introduced at a lower altitude for the most extreme geometries (SZA $89°$, VZA $85°$ and $89°$). This can be attributed to the sensitivity of percent error to small numerical errors when the compared values are small, as the flattened atmosphere has reduced the box-AMFs to near zero near the surface due to the increased path lengths.

### 4.3 Timing

Table 2 contains timing results for the comparisons presented in Sections 4.1 and 4.2. The main difficulty in comparing timing is that computation times scale differently; SASKTRAN-MC scales with number of lines of sight, SASKTRAN-HR scales with number of AMF layers, and SASKTRAN-DO scales weakly with the number of lines of sight. The timing as a function of lines of sight and AMF layers is shown in Table 2, but to give a more practical comparison the total time required for an example of a full AMF table is also estimated. The example used here is taken from the tables described below in Section 5; an

aerosol-free clear sky table containing $400$ scenes (10 SZA, 10 surface albedo, and 4 surface pressures), 49 lines of sight per scene (7 viewing zenith angles and 7 azimuth angles), and 100 AMF layers ($500\,\mathrm{m}$ spacing up to $50\,\mathrm{km}$).

SASKTRAN-DO is going to be the fastest option for most applications, unless two- or three-dimensional analysis is required. Currently SASKTRAN-DO is only configured to multithread over wavelength, and these are single wavelength calculations, so the times will be improved further when multithreading over other parameters is implemented. Monte Carlo is clearly not

suitable for full table generation; even a modest precision of $1\%$ would require on the order of 1200 hours of computation. The time required for precision $N$ is proportional to $N^{-1/2}$, so, for example, to achieve $0.5\%$ the computation time would increase

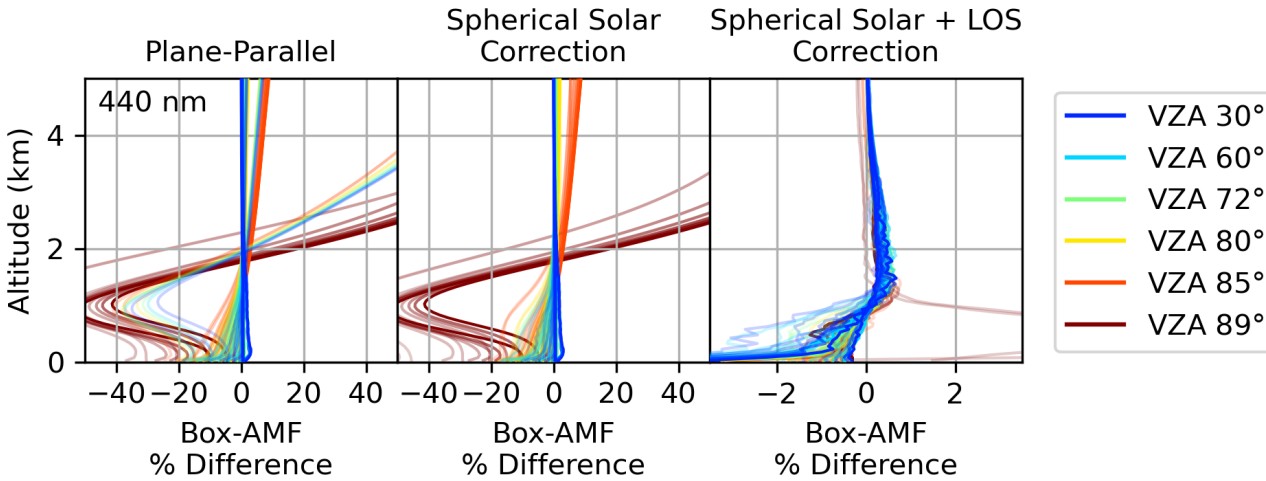

**Figure 5.** Box-AMF comparisons between SASKTRAN-MC and SASKTRAN-DO in an atmosphere that has been effectively flattened by a factor of 10. The increased noise is due to fewer photon paths (1e6) being traced. Note that the x-axis scale changes in the rightmost panel.

**Table 2.** Timing comparisons based on an Intel Core i7-6700 CPU at $3.4\,\mathrm{GHz}$ with $16\,\mathrm{GB}$ of RAM on Windows 10 using 8 threads. $L$ is the number of lines of sight per scene, and $A$ is the number of AMF layers. The timing estimates for a full table assume 400 scenes with 49 lines of sight and 100 AMF layers. DO is currently only configured to multithread over wavelength; these single-wavelength DO times will be improved when multithreading over other parameters is implemented. HR AMFs are currently computed via finite-difference approximation; when HR is linearized the dependence on the number of AMF layers will be greatly reduced.

| Method | Settings | Time per scene $(s)$ | Time per table $(h)$ |
|---|---|---|---|
| MC | to $1\%$ precision | $220L$ | 1200 |
| HR | 144 incoming, 1 profile | $1.1A$ | 12 |
| HR | 792 incoming, 1 profile | $5.8A$ | 12 |
| HR | 144 incoming, 9 profile | $11A$ | 130 |
| HR | 792 incoming, 9 profile | $70A$ | 780 |
| DO | No LOS correction | $0.209 + 0.011L$ | 0.083 |
| DO | LOS correction | $0.517 + 0.088L$ | 0.54 |



by a factor of 4. SASKTRAN-HR AMFs will be sped up in the future when full linearization is implemented, removing the need for redundancy in the finite difference approach.

## 5 SASKTRAN AMF Tables

SASKTRAN-derived box-AMF profiles have been used by Griffin et al. (Griffin et al., 2019, 2021) from Environment and Climate Change Canada (ECCC) to analyze TROPOMI measurements over North America. The first application compared TROPOMI data with in-situ aircraft, in-situ ground-based, and remote ground-based $NO_2$ measurements over the Canadian oil sands, improving agreement through use of regional, higher density retrieval inputs. The second application examined $NO_2$ retrievals over North American forest fires from 2018 and 2019, this time improving agreement between TROPOMI and aircraft measurements in part by using box-AMF profiles that explicitly account for aerosol content. The AMF lookup tables that were generated for these projects are available at https://zenodo.org/record/6629418, and tools for computing box-AMFs with SASKTRAN can be found at https://arg.usask.ca/docs/skdoas/.

Table 3 outlines the parameter space for the AMF lookup tables. All tables were done on a $500\,\mathrm{m}$ vertical grid. The original table spanned $0\,\mathrm{km}$ to $16\,\mathrm{km}$ at a wavelength of $440\,\mathrm{nm}$ for use with tropospheric $NO_2$. Subsequent tables added one ore more of the following features: extending up to $50\,\mathrm{km}$, adding another wavelength at $330\,\mathrm{nm}$, and adding explicit aerosol layers. Further modifications, such as ozone parameterizations, non-Lambertian surface reflection, or even parameterizations accounting for horizontal inhomogeneity, are certainly possible. SASKTRAN-HR was used for all of the tables up to this point, with nominal settings similar to those used for the middle panel of Figure 1, which show agreement of about $1\,\%$ with SASKTRAN-MC; multiple diffuse profiles and high density incoming grids were deemed unnecessary for the range of viewing geometries included in the tables. At the time the spherical corrections for SASKTRAN-DO were not implemented; now that they are available, future iterations can utilize this model.

## 6 Two-Dimensional Sensitivity Study

In the following study, a potential application of SASKTRAN-HR's capacity for horizontally inhomogeneous atmospheres is demonstrated. A two-dimensional analysis is performed for a simplified TEMPO-like winter $NO_2$ measurement over the Canadian oil sands, a region of interest near the northern extent of the field of regard of TEMPO. A scenario with significant horizontal variation, both in the $NO_2$ and the the surface albedo, is constructed, and the total AMF is computed accounting for this variation, and again while neglecting it. The difference quantifies the consequences of the assumption of horizontal inhomogeneity that one-dimensional analyses are built upon.

The scenario for this study was inspired by simulated and measured data in order to ensure realistic values, but was greatly simplified in order to keep interpretation manageable. Two $NO_2$ profiles, one with surface pollution and one without, were selected from the scene shown in Figure 6, taken from the global tropospheric chemistry simulations by Hu et al. (Hu et al., 2018). The surface reflection is assumed to be Lambertian for simplicity, with an albedo of $0.8$ in the south (approximating



**Table 3.** Parameters for SASKTRAN-HR AMF lookup tables used by ECCC. Aerosol optical depth and layer height were only used for the wildfire study (Griffin et al., 2021), not the oil sand study (Griffin et al., 2019).

| Parameter | Table | Values |
|---|---|---|
| Solar zenith angle ($°$) | Both | 0, 30, 50, 60, 65, 70, 73, 76, 78, 80 |
| Viewing zenith angle ($°$) | Both | 0, 30, 50, 60, 65, 70, 72 |
| Azimuth angle difference ($°$) | Both | 0, 30, 60, 90, 120, 150, 180 |
| Surface albedo | Clear | 0.00, 0.03, 0.06, 0.09, 0.12, 0.20, 0.30, 0.50, 0.75, 1.00 |
| Surface pressure (Pa) | Both | 6e4, 8e4, 9e4, 1e5 |
| Cloud top albedo | Cloudy | 0.8 |
| Cloud top pressure (Pa) | Cloudy | 2e4, 4e4, 6e4, 8e4, 9e4 |
| Aerosol optical depth | Clear | 0.00, 0.03, 0.10, 0.20, 0.50, 1.00, 2.00, 3.00 |
| Aerosol layer height (km) | Clear | 1.0, 2.0, 3.0, 4.0 |
| Wavelength (nm) | Both | 440 |

the reflectivity of snow) and $0.2$ in the north. These values were selected as a rough representation of the Moderate Resolution Imaging Spectroradiameter (MODIS) data shown in Figure 6, which is the nadir BRDF-adjusted reflectance (NBAR) from

band 3, which spans $459\,\mathrm{nm}$ to $479\,\mathrm{nm}$ (Schaaf and Wang, 2015). Both scenes were taken from December 15, 2013, 18:00 UTC.

     The simplified two-dimensional scenario is illustrated in Figure 7, showing the polluted $NO_2$ profile over snow in the south, and the unpolluted $NO_2$ profile over a lower albedo surface in the north. It also shows line of sight and the direct sun beam for the TEMPO-like viewing geometry on a winter day at approximately $54°$ latitude, with a viewing zenith angle of $62°$ and a

solar zenith angle of $78°$.

     The first step was to compute two-dimensional box-AMFs, with and without the horizontally varying surface albedo. They were computed across approximately $200\,\mathrm{km}$ horizontally and up to $5\,\mathrm{km}$ in altitude, covering most of the sensitivity to horizontal variability for this scenario. The two-dimensional box-AMFs, along with their corresponding one-dimensional box-AMFs, are shown in Figure 8. As expected, neglecting the high reflectivity to the south results in an underestimation of the

box-AMFs, and therefore an underestimation of the measurement sensitivity. Note that each column is the approximate width of four TEMPO pixels at this latitude.

     The second step is to compute total AMFs by taking weighted averages of the box-AMF values using $NO_2$ concentrations as weights (see Equation 30). Figure 9 shows the partial slant columns $S_{ij}$, which are an intermediate quantity in this computation. Note that the sum of all $S_{ij}$ returns the total slant column, so that Figure 9 is a visualization of the distribution of the origin



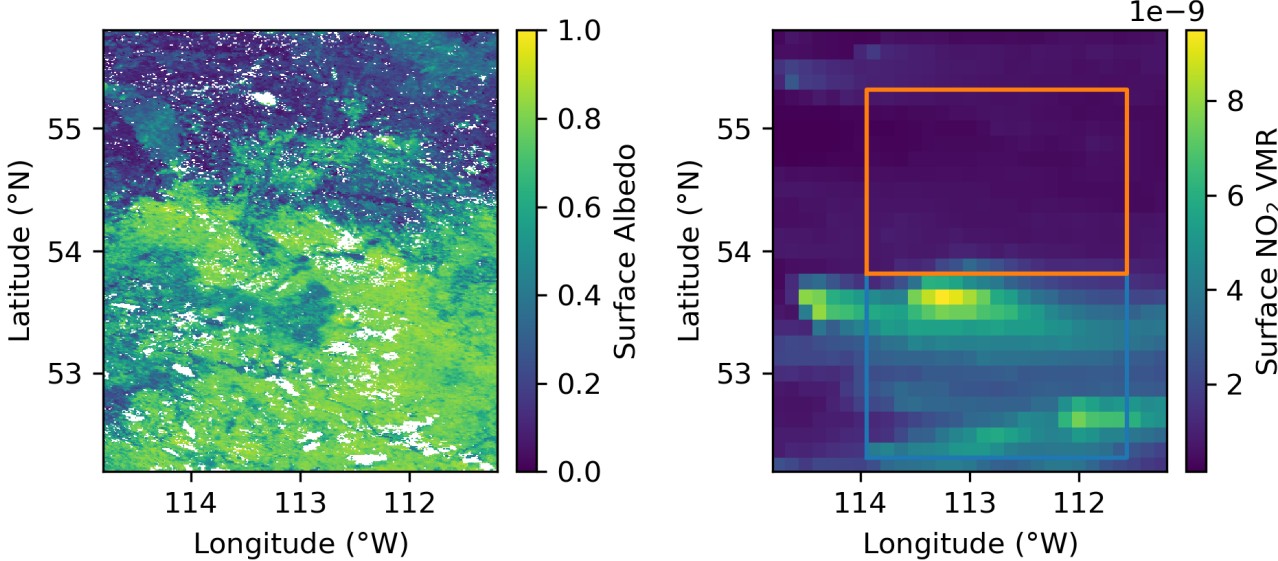

**Figure 6.** Surface albedo and surface $NO_2$ data used to justify the simplified scenario for the two-dimensional study. The albedo is the nadir BRDF-adjusted reflectance MODIS data (Schaaf and Wang, 2015) and the $NO_2$ is simulated (Hu et al., 2018). The polluted and unpolluted $NO_2$ profiles are averages from the blue and orange box respectively.

of the measured signal. The enhancement in signal originating from the lowest layers south of the ground pixel is evident, particularly when both the albedo and the $NO_2$ are increased in the bottom right panel.

First, consider the total AMF for a scenario with variable surface albedo as described in the original scenario, but with horizontally homogeneous $NO_2$. Combining the box-AMFs computed with uniform surface albedo with the horizontally homogeneous $NO_2$ (see Figure 9, top-left) results in an AMF of 1.21; this is what a one-dimensional analysis would return.

Combining the box-AMFs computed with the variable surface albedo with the horizontally homogeneous $NO_2$ (see Figure 9, bottom-left) results in an AMF of 1.64. Neglecting the change in surface albedo for this scenario results in underestimating the total AMF by 26 %.

Second, consider a scenario with horizontally inhomogeneous $NO_2$ as described in the original scenario above, but with uniform surface albedo. The one-dimensional analysis of this scene is identical to the previous, resulting in a total AMF of 420 1.21. The two-dimensional analysis combines the uniform surface albedo box-AMFs with the true $NO_2$ field (see Figure 9, top-right), resulting in a total AMF of 1.91. Neglecting the horizontal change in $NO_2$ for this scenario results in underestimating the total AMF by 37 %.

Finally, in the same way consider the original scenario with horizontal variation in both surface albedo and $NO_2$; here (see Figure 9, bottom-right) the true total AMF is 4.05, meaning that neglecting both horizontal changes together results in 425 underestimating the total AMF by 70 %.



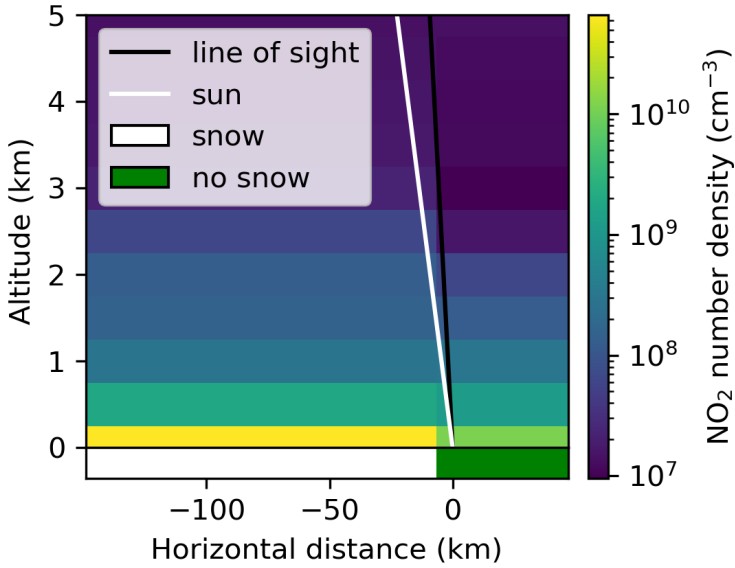

**Figure 7.** Scenario used for the two-dimensional sensitivity study. Shown is a TEMPO-like measurement of an unpolluted scene over a surface albedo of 0.2, but heavy pollution over snow (surface albedo 0.8) is found to the south.

**Table 4.** Effects of neglecting horizontal variation on total AMF. The distance of the ground pixel from the change in albedo/$NO_2$ is given in TEMPO pixels by $p$ and in distance by $x$. The column headings for total AMFs indicate which quantity has horizontal variation in the simulated scene, and the percent difference quantifies the error when this variation is ignored when computing the AMF.

| Position | | | Total AMF | | |
|---|---|---|---|---|---|
| $p$ | $x$ (km) | Constant | Albedo (% Diff) | $NO_2$ (% Diff) | Both (% Diff) |
| 2 | 6.5 | 1.22 | 1.67 (-27.3) | 1.93 (-37.0) | 4.17 (-70.8) |
| 6 | 19.6 | 1.19 | 1.45 (-17.9) | 1.56 (-23.4) | 2.79 (-57.3) |
| 14 | 45.7 | 1.18 | 1.28 (-7.9) | 1.31 (-9.7) | 1.78 (-33.8) |
| 30 | 97.8 | 1.16 | 1.18 (-1.1) | 1.20 (-2.9) | 1.30 (-10.4) |

These results are somewhat severe due to the close proximity of the ground pixel to the sudden jump in surface albedo and $NO_2$; perhaps the more interesting question is how far away from such a feature does the ground pixel need to be before the effects become negligible. Table 4 summarizes the results of the same analysis while the ground pixel is moved progressively further north. With either feature on its own, errors on the order of $10\%$ can be found at a distance of nearly $50\,\mathrm{km}$ away; with both features combined $10\%$ errors can be found at a distance of nearly $100\,\mathrm{km}$.






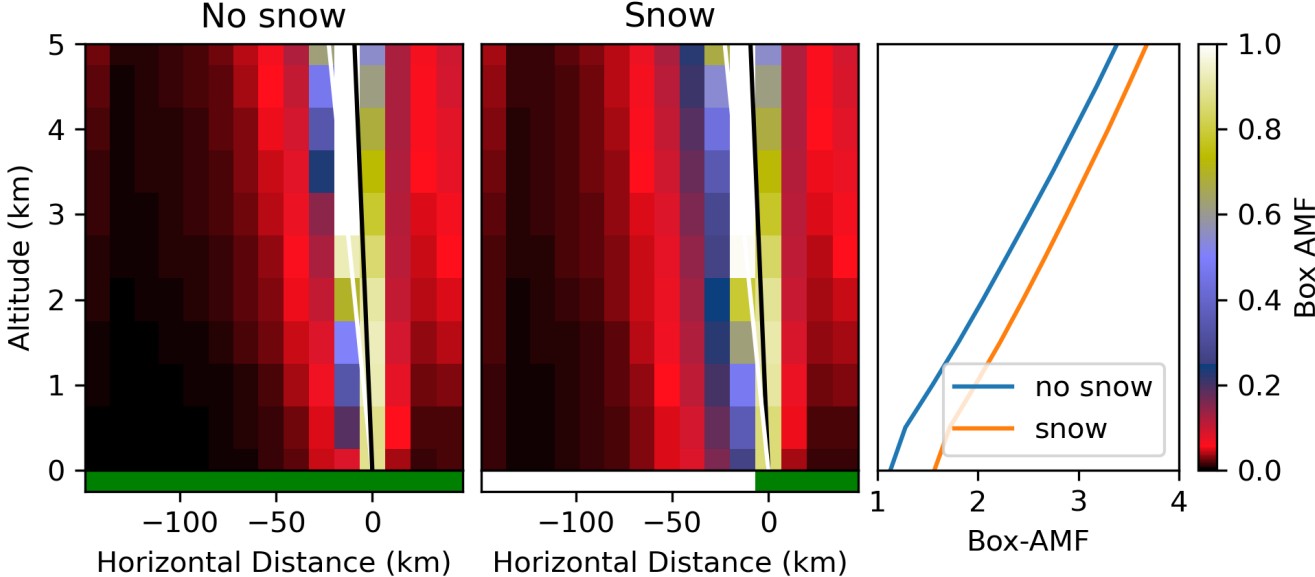

**Figure 8.** Two-dimensional box-AMFs with and without a region of high reflectivity south of the ground pixel. The outermost columns represent the contribution from the entire field beyond what is shown here; this is why these box-AMFs increase slightly while the trend is clearly decreasing. The sum of all two-dimensional box-AMFs at a given altitude recovers the traditional one-dimensional box-AMF, shown on the right.

The usefulness of such a strategy for accounting for horizontal variations could be evaluated in an operational setting by comparing these errors with other sources of error. AMF errors are already quite high, for example the typical errors of $NO_2$ AMFs for the Tropospheric Monitoring Instrument (TROPOMI) are estimated to be $15\%$ to $25\%$, but can easily exceed $50\%$ under the right circumstances (van Geffen et al., 2022). The above analysis suggests that horizontal variations could easily

contribute errors on the order of $15\%$ to $25\%$, but such occurrences are spatially sparse due to the requirement of large horizontal gradients. This analysis also does not account for the horizontal resolution of the input $NO_2$ field and albedo. The horizontal analysis of the radiative transfer implies a certain horizontal resolution to the measurement; if the input products match this resolution, such errors would be reduced.

This approach is not currently feasible on a large scale due to the large computational load, especially for the volume of data

supplied by the normal operation of an instrument like TEMPO. Primary obstacles include high computation times required for two- or three-dimensional fields, difficulty in parameterizing such fields for a lookup table, and the accuracy and availability of prior trace gas fields at such high horizontal resolutions. Using this approach for smaller scale studies or campaigns is much more feasible. For example, using it only for localized analysis of winter scenes containing select industrial or urban regions would filter out much of the data volume while maximizing occurrences of large horizontal gradients. It could also

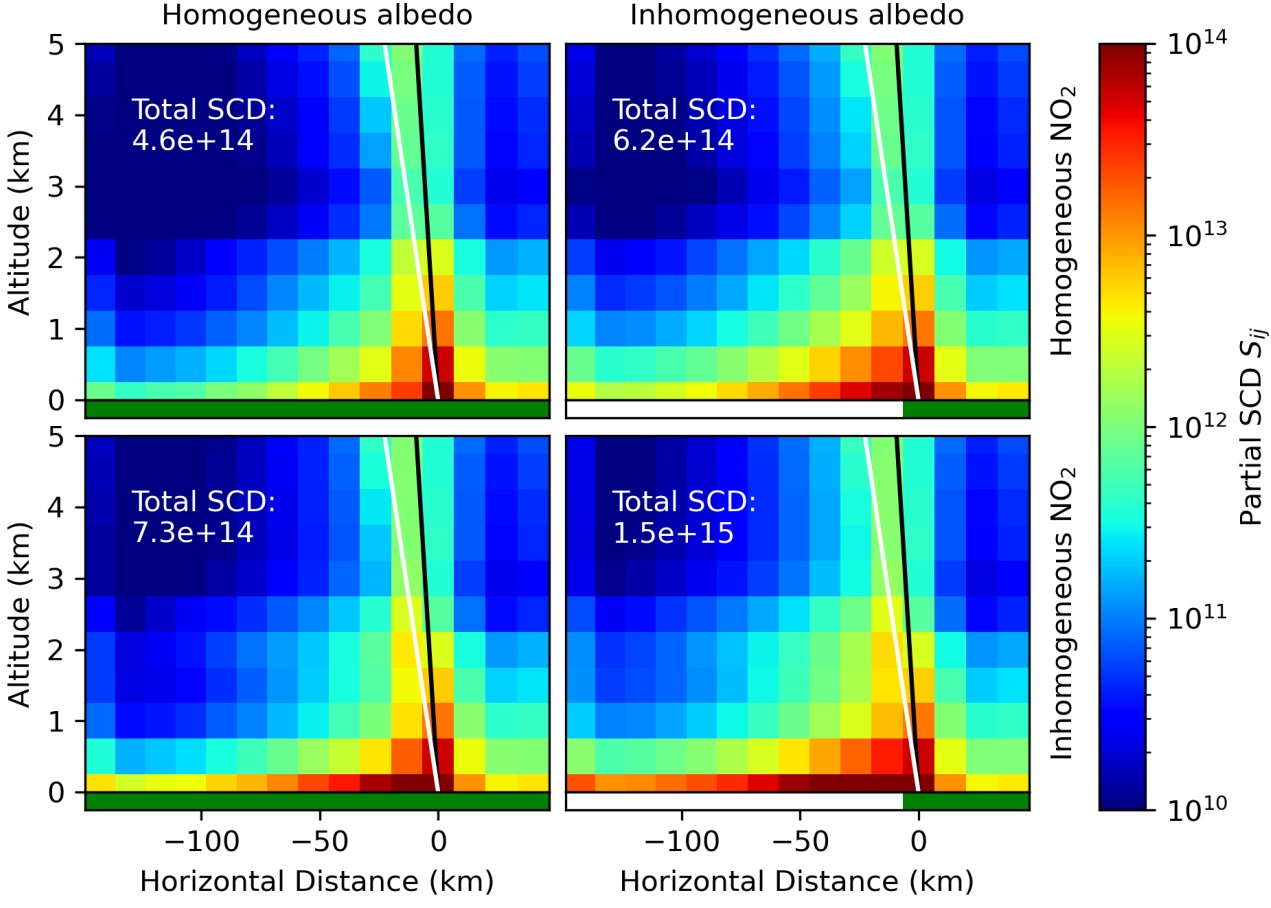

**Figure 9.** Partial SCD distribution for scenes with horizontally homogeneous and inhomogeneous surface albedo and $NO_2$. The sum of each pixel returns the total SCD that would be measured by the instrument.

be used effectively for special observations from TEMPO that focus on events producing large gradients, such as forest fire observations with reduced revisit time (Zoogman et al., 2017).

Computation times could be greatly reduced by fully linearizing SASKTRAN-HR, which would eliminate redundancy in the current finite-difference approach; this upgrade is to be implemented within the next few years. Another potential improvement is separating contributions from the line of sight and single scatter paths from the diffuse multi-scatter field, removing sharp

features and permitting a large reduction in horizontal resolutions.

There are many potential alternative applications of a multi-dimensional AMF field to satellite measurements. For example, dependence on assumed $NO_2$ fields could potentially be reduced by analyzing multiple pixels simultaneously, utilizing the data from adjacent pixels which would otherwise be ignored. Such a method could be particularly effective for a localized analysis combining satellite measurements with in-situ measurements. As another example, multi-dimensional AMF fields would add



value as part of chemical data assimilation. Furthermore, they could be used to estimate an albedo and geometry dependent
horizontal averaging kernel, characterizing the contribution of radiative transfer to the true horizontal resolution that is being
measured.

## 7 Conclusions

SASKTRAN, originally designed for limb measurements, has been upgraded for use in nadir applications. Air mass factor
computation has been added to the Monte Carlo method (SASKTRAN-MC) which serves as an important validation tool
for the successive orders (SASKTRAN-HR) and discrete ordinates (SASKTRAN-DO) methods. SASKTRAN-DO has been
equipped with spherical corrections which make the method feasible at extreme geometries. Air mass factors computed with
all three methods were computed and found to be in good agreement. Agreement between SASKTRAN-HR and SASKTRAN-
MC for moderate geometries was found to be on the order of $1\%$ with default settings, and could be brought as low as $0.4\%$ by
increasing the resolution of the downwelling and near-horizontal radiance field. Agreement on the order of $2\%$ can be achieved
for extreme geometries, requiring the use of multiple diffuse profiles. Agreement between SASKTRAN-MC and SASKTRAN-
DO (with spherical solar and line of sight corrections) was found to be on the order of $2\%$ for most geometries, with a distinct
feature at mid-altitudes under all sun positions and viewing geometries due to the plane-parallel approximation in the multiple
scattering.

SASKTRAN-HR is equipped to handle two- and three-dimensional features, providing a deterministic alternative to Monte
Carlo for applications calling for horizontal analysis. For example, increased horizontal interference would be expected in the
presence of strong horizontal gradients in surface albedo (e.g. light or variable snow, coastlines) or in trace gas concentrations
(e.g. urban centers, industrial emitters, forest fires). SASKTRAN-HR was used to perform a sensitivity analysis on a simulated
TEMPO scene over the Canadian oil sands, near the northern extent of its field of regard. The surface albedo was made to
transition from $0.2$ to $0.8$ and the $NO_2$ field from unpolluted to polluted at varying distances from the ground pixel. The
two-dimensional distribution of the light path and the measured $NO_2$ signal were calculated and visualized, and the impact of
neglecting horizontal changes was investigated. Errors on the order of $10\%$ were estimated at distances up to $50\,km$ with one
of these features present, and at distances up to $100\,km$ with both.

This study demonstrates that error due to horizontal variability is significant for a TEMPO-like instrument in the presence
of sufficiently large horizontal gradients in surface albedo or trace gas concentration. However, accounting for it on a large
operational scale is not advised due to computational requirements and the sparsity of such gradients. Localized analysis of
scenes that are expected to contain large gradients stand to benefit the most, such as winter scenes containing industrial or urban
regions of interest or TEMPO special observations of events like forest fires. Future work includes increasing the computational
efficiency of the multi-dimensional radiative transfer and exploring the effectiveness of non-traditional retrieval methods, such
as simultaneous analysis for groups of adjacent pixels, explicit combination with other measurements sources, or injection into
a climate-chemistry model.



*Code and data availability.* Code and data is available at https://doi.org/10.5281/zenodo.6629417 (Fehr, 2023)

*Author contributions.* Lukas Fehr implemented the air mass factor computation upgrade for SASKTRAN-MC, performed the one-dimensional air mass factor comparisons and the two-dimensional study, and wrote the manuscript under the supervision of Adam Bourassa and Doug

Degenstein. Daniel Zawada implemented the spherical corrections for SASKTRAN-DO and provided radiative transfer consultation. Chris McLinden and Debora Griffin provided air mass factor consultation.

*Competing interests.* The authors declare that they have no conflict of interest.



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
