# Peer review of "Spherical Air Mass Factors in One and Two Dimensions with SASKTRAN 1.6.0"

_Geoscientific Model Development, 2023_

## Author Response (AR1)

| section | location | ref # | referee comment | my response |
|---|---|---|---|---|
| | | 1 | More details should be given about the derivation of the AMF in the specific cases of HR and DO. | Section 3.5 connects box-AMFs to derivatives in HR and DO, which are discussed in 2.1 and 2.3. I added a bit more context to 2.1 and 2.3. |
| | | 1 | It is interesting to see the effect of wavelength changes on the differences HR/MC and HR/DO. For these comparisons, I would suggest including aerosols layers in the model atmospheres. It is needed, since, in section 6, aerosols are considered in the sensitivity study. | Aerosols are included in the tables described in section 5, but are not included in the sensitivity study in section 6. This was unclear because table 3 (which mentions aerosols) belongs to section 5 but was placed after the start of section 6. I have made this more clear in the introduction to section 6. Adding aerosols to sections 4 and 6 would be a logical next step for these studies, but care would have to be taken to keep the number of configurations manageable. |
| 2.1 | Line 85 | 1 | it could be recalled that the radiance is the sum over n of the radiances scattered n times (even if it seems obvious). | done |
| 2.1 | Line 101 | 1 | I agree with the fact that, when the state of the atmosphere depends only on the SZA and the altitude, only these two spatial coordinates need to be considered. However, I disagree with the reason you give: there is no rotational symmetry around the solar direction, since there is an other privileged direction: the vertical (the grounds acts as a radiance source when the albedo is nonzero). | I have now further specified that the surface reflectivity must have rotational symmetry for the statement to be correct. |
| 2.1 | Lines 111-112 | 1 | if the built-in weighting function are analytical, could you mention it ? | done |
| 2.1 | Lines 111-112 | 1 | could you precise what approximations make built-in AMF unsuitable for precise AMF computations ? | I have reworded the sentence so that the reader is more clearly pointed to the reference for details regarding the approximations. |
| 2.1 | Table 1 | 1 | I like the idea of recalling the definitions. However, they should be accompanied by figures to make the geometry more clear. | I've added an illustration for the HR settings. |
| 2.2 | Line 121 | 1 | Could you define mathematically x_n ? It is a function s -> (z,lon,lat)(s) ? | This is meant to be general. I have reworded it to make this more clear, and also included an example definition. |
| 3.1 | Equation 8 | 1 | Equation 8 defines the AMF for the MC method only, it should be specified. For definition for the other methods should | Equation 8 is general, I've made it more clear now what applies to each method. |
| 3.2 | Line 189 | 1 | "equivalent (up to a sign)" should be replaced by "equal to the opposite of". | done |
| 3.2 | Equations 10 and 11 | 1 | Wouldn't it be possible to homogenize the notations between equations 10 and 11 ? (k(z)+ epsilon * phi(z) in 10, kbar + Delta_k(z) in 11 ? | done |
| 3.3 | Equation 15 | 1 | quation 15 rather defines an effective layer geometrical thickness A effective layer height would be \int_{0}^{H} z\phi_i(z) dz | done |
| 3.4 | | 1 | To be as general as possible, it would have been preferable to write the equations in the 3D case with longitude and latitude. | Agreed, I have made this section more general. |
| 3.5 | Lines 259-260 | 1 | is the assumption that the change in sigma(z) is negligible valid whatever the molecules considered ? This point should be detailed in the paper. | Rather than assert this is true for all cases, I have made it clear that this is an assumption I am making for this derivation. |

| | | | | |
|---|---|---|---|---|
| 3.5 | Equation 35 | 1 | For HR method, is this equation applied for the radiance at every order of scattering ? For their sum ? | This equation uses the total radiance I. |
| 4.1 | Lines 290-291 | 1 | what is the complete list of molecules considered ? No H2O ? | NO2 and O3 is the coplete list of absorbing species. |
| 4.1 | Line 295 | 1 | What is a moderate geometry ? | There isn't a precise definition but I've added more context. |
| 4.1 | Lines 300-301, 306-306 | 1 | Is there a reason why there are more downward facing than upward facing directions? | It was found that increasing the upward facing resolution did not bring appreciable improvement. |
| 4.1 | Line 307 | 1 | Precise what is an extreme geometry. | There isn't a precise definition but I've added more context. |
| 4.1 | Figure 1 | 1 | In figure 1, right graph. What is the meaning of a (e.g. "a 0.05") ? | done |
| 4.2 | Figure 4 | 1 | In figure 4, the effect of the SZA is not very clear: the difference in transparency is difficult to distinguish. I would advise less SZA, with curves for every SZA distinguished otherwise (dashed lines, ...) | I have made figures 4, 5, and 6 more readable and precise. |
| 4.2 | Line 336 | 1 | "small zenith angle" :  SZA or VZA | done |
| 4.2 | Lines 342-348 | 1 | Changing the radius is a simpler way to change the sphericity. Could you try it ? | Changing the radius is currently more difficult in our code. |
| 6 | | 1 | A 3D study would have been more realistic. | Agreed, but also more computationally expensive and complex to interpret, we decided this was a good starting point. |
| 6 | | 1 | What order of scattering (max  value of n in eq 1) was | I added a comment in section 4.1. |
| 6 | Table 3 | 1 | what are the nature of  cloud and aerosol particles ? SSA ? What size distributions where used ? A plot of their phase function would be interesting. Where the aerosol / cloud properties dependent on the altitude ? | I have added details about the size distribution and the altitude dependence. |
| 6 | Figure 6 and 7 | 1 | In figure 6, NO2 VMR, is used, in figure 7, NO2 number density. You should use the same quantity in both figure. | done |
| 6 | | 1 | Could you plot the 2 NO2 profiles vertical chosen  (cle | done |
| 6 | Figure 8 | 1 | Figure 8 should be completed, to have the same graph for the four cases shown in figure 9 (homogeneous/inhomogeneous albedo, homogeneous/inhomogeneous NO2 ). | The same box-AMFs were used for both NO2 scenarios, since the change in box-AMFs with absorber profile is negligible. I have added a note to the text discussing Figure 8. |
| 6 | Figure 8 | 1 | Figure 8, graph at the right: could you plot both the 1D and the sum of 2D box-AMF ? | done |
| 6 | Figure 9 | 1 | Figure 9: could you precise the unit of total and partial SCD ? | done |
| | Line 112 | 1 | AMF acronym should be defined here (first occurrence), not in line 163 | The acronym is defined on line 24, but I have now removed the redundant definition on line 163. |
| | Table 2 | 1 | "9 profile" => "9 profiles" (2 occurrences) | done |
| | Figure 6 | 1 | In figure 6, the blue box is difficult to distinguish in the right graph. A change of color (maybe using dashed line) would make it easier to read. | done |
| | Figure 6 | 1 | In figure 6, some white patches appear in the left graph. You should specify what they mean (white = no data ?) | done |
| | | 2 | I feel that some simple drawings or something similar illustrating the various AMFs in section 3 would make the material even more accessible to readers less familiar with the topic. | I've added a couple AMF illustrations to section 3. |

| | | | |
|---|---|---|---|
| | 2 | The calculations are carried out on scenarios with increasing complexities. It is not entirely clear to me whether aerosols have been included in the atmosphere. Reading the bulk of the paper I've got the impression they have not, but Table 3 lists aerosol optical depth. Do the results presented in Figures 1-4 include the impact of aerosols? Due to their phase function being distinctly different from those of molecules the size of the area of pixels with varying surface reflectances that impact radiances in the viewing direction from a pixel inside the area would be different depending on whether or not the atmosphere includes aerosols. | The AMFs computed in section 4 and the 2D study in section 6 do not have aerosols, only the tables described in section 5. I have made this more clear. Adding aerosols would be a logical next step for these studies, but care would have to be taken to keep the number of configurations manageable. |
| Line 35 | 2 | Define VCD here since I think this is the first time it is mentioned.Line 43: TEMPO was already launched. It was launched on April 7, 2023 so the authors may want to revise the sentence. | done |
| Line 75 | 2 | Perhaps it would be better to write "modelling horizontal inhomogeneities" instead of "modelling horizontally inhomogeneities". | done |
| Table 1 | 2 | Do "Layers" and "Streams" also apply to the DO method in addition to MC? | They apply exclusively to DO, I have fixed the typo. |
| Line 131 | 2 | It may be useful to define box-AMF here or point the reader to the place where it is defined. | I have made sure box-AMF is defined before it is used. |
| Equation 10 | 2 | Please define epsilon. I did not see it defined before this equation, but it is possible I missed it. | I have changed the notation, but also added a definition for the term in question. |
| Figure 1 | 2 | In the caption, you may want to add "... and surface albedos, a, 0.05 and 0.8". This would inform the reader what "a" in the legend is without searching the text that comes after the figure (at least it is the case in the manuscript). | done |
| Lines 298-301 | 2 | Would it possible to also include a cartoon showing the arrangement described in the text? | done |
| Lines 374-375 | 2 | This should say "... added one or more ...". | done |
| Line 388 | 2 | Should this read " ... homogeneity ..." instead of " ... inhomogeneity ..."? | yes, fixed |

---

## Author Response (AR2)

Here are 2 final (very minor) suggestions from referee 1 before the paper is accepted:

Section 3:
Line 104: "has rotational symmetry" I think you mean lambertian surfaces. Could you specify it ?

Response:

We have clarified the statement in the paper. The new statement does not imply that Lambertian surfaces are a requirement for the dimensional reduction, since a more general class of BRDFs (of which Lambertian surfaces are a subset) permit it, but it does not get into the details that precisely define this class since this is not crucial for the paper.

Previous statement:

The radiation field is five-dimensional, with three spatial and two directional coordinates, but due to rotational symmetry around the solar direction the spatial dimensions can be reduced to two (SZA and altitude) when the atmosphere is a function of only altitude and SZA and when the surface reflectivity has rotational symmetry.

New statement:

The radiation field is five-dimensional, with three spatial and two directional coordinates, but symmetries around the solar zenith reduce the dimensionality to 4 for certain atmospheric and surface configurations, including common cases such as a horizontally homogeneous spherical shell atmosphere and Lambertian surface.

Figure 7: there is no legend for the different types of lines on the right of the figure.
Could you add it, as it has already been done in figure 6 ?

Response:

We have added the requested legend to Figure 7. We have also altered the spacing of the equivalent legends in Figures 4 and 6 so they are all consistent.